# Ha-Ras^V12^-Induced Multilayer Cellular Aggregates Is Mediated by Rac1 Activation Rather Than YAP Activation

**DOI:** 10.3390/biomedicines10050977

**Published:** 2022-04-23

**Authors:** Li-Ying Wu, Chia-Lin Han, Hsi-Hui Lin, Ming-Jer Tang

**Affiliations:** 1Department of Physiology, College of Medicine, National Cheng Kung University, Tainan City 70101, Taiwan; abigailwu8312@gmail.com (L.-Y.W.); chia-lin.han@cytiva.com (C.-L.H.); 2International Center for Wound Repair and Regeneration, National Cheng Kung University, Tainan City 70101, Taiwan

**Keywords:** Ha-Ras^V12^, Caveolin-1, YAP, myosin IIB, Rac, cell aggregates

## Abstract

We demonstrate that Ha-Ras^V12^ overexpression induces the nuclear translocation of Hippo effector Yes-associated protein (YAP) in MDCK cells via the hippo-independent pathway at the confluent stage. Ha-Ras^V12^ overexpression leads to the downregulation of Caveolin-1 (Cav1) and the disruption of junction integrity. It has been shown that the disruption of actin belt integrity causes YAP nuclear translocation in epithelial cells at high density. Therefore, we hypothesized that Ha-Ras^V12^-decreased Cav1 leads to the disruption of cell junction integrity, which subsequently facilitates YAP nuclear retention. We revealed that Ha-Ras^V12^ downregulated Cav1 through the ERK pathway. Furthermore, the distribution and expression of Cav1 mediated the cell junction integrity and YAP nuclear localization. This suggests that the downregulation of Cav1 induced by Ha-Ras^V12^ disrupted the cell junction integrity and promoted YAP nuclear translocation. We further indicated the consequence of Ha-Ras^V12^-induced YAP activation. Surprisingly, the activation of YAP is not required for Ha-Ras^V12^-induced multilayer cellular aggregates. Instead, Ha-Ras^V12^ triggered the ERK-Rac pathway to promote cellular aggregate formation. Moreover, the overexpression of constitutively active Rac is sufficient to trigger cellular aggregation in MDCK cells at the confluent stage. This highlights that Rac activity is essential for cellular aggregates.

## 1. Introduction

Ras is a small GTPase containing three isoforms, including Harvey-Ras (Ha-Ras), Kirsten-Ras (K-Ras), and N-Ras. More than 30% of human cancers are driven by one or multiple point mutations of the Ras gene [1]. The mutation of Ras at position 12 from glycine to valine results in a GTP-bound state, which is constantly active to trigger the downstream signaling for cell proliferation, cell invasion, cell migration, and cytoskeleton organization, so as to cause cell tumorigenesis [2,3]. The Ha-Ras^V12^ mutant has been observed in many human cancers, especially in head and neck squamous cell carcinoma and bladder cancer [4].

Around 90% of human cancer is from epithelial tissue. Oncogenes, such as Ras, promote the regulation of the epithelial–mesenchymal transition (EMT) in epithelial cells [5]. To maintain the properties and function of epithelial cells, cell junctions play an important role. In epithelial cells, the intracellular force is provided by actin and actomyosin cytoskeletons, which form stress fibers and adhesion belts on cell–cell contacts and cell–ECM sites [6]. In epithelial cells, there are two types of heavy chains in myosin II, including myosin IIA and myosin IIB. They also have diverse functions in the regulation of cell morphology, cell polarity, and cell orientation [7]. Myosin II has also been reported to contribute to cancer progression. The absence of myosin IIB induces cell proliferation and anti-apoptosis through the upregulation of nuclear YAP in epithelial cells [8].

YAP, a transcriptional cofactor regulated by the Hippo pathway, was first discovered in Drosophila and mammals. The disturbance of the Hippo–YAP pathway results in tissue and body overgrowth [9,10]. Nevertheless, a recent report revealed that the Hippo pathway is not the only signaling pathway that regulates YAP activity. YAP nuclear localization and activation are also regulated by ECM stiffness and Rho-mediated tension of actin stress fibers [11,12]. During cancer progression, EMT triggers cancer cells’ aggressive behavior, such as proliferation, invasion, and migration [13]. The activation of YAP plays an essential role in EMT progression in mammary epithelial cells. A recent report demonstrated that constitutive active YAP-overexpressed MDCK cells were extruded from the normal MDCK monolayer and formed multilayer cellular aggregates [14]. Moreover, the activation of YAP has been found in several cancer types, such as liver, breast, colon, skin, lung, and ovarian [15,16,17].

Even though the relationship between cancer and cell extrusion is not clear, Rac, another small GTPases, has been reported to be involved in cancer progression. The inhibition of Rac has been reported to prevent cell extrusion; however, the underlying mechanism is not well understood [18]. Rac is well known in the regulation of lamellipodia and membrane ruffles via the WASP-family verprolin-homologous protein 1 (WAVE1), which regulates the actin cytoskeleton [19]. Rac has been proven to be involved in the regulation of the cell junction integrity. However, the effect of Rac on adherens junctions is controversial. Some studies have revealed that the overexpression of both constitutively active and dominant-negative Rac promotes the disassembly of the apical junctional complex [20]. Another study demonstrated that the adherens junction complex, E-cadherin–catenin complex, degrades in Ha-Ras^V12^Rac (Ha-Ras constitutively active form plus Rac wildtype)-overexpressing cells and Ha-Ras^V12^Rac^V12^ (Ha-Ras constitutively active form plus Rac constitutively active form)-overexpressing cells, which can be prevented in Ha-Ras^V12^ plus Rac^N17^ (Ha-Ras constitutively active form plus Rac-dominant negative form) overexpression cells [21]. The contribution of Rac to the cell junction integrity and cell extrusion is still not well known.

Our previous study also demonstrated that the overexpression of Ha-Ras^V12^ downregulated Cav1 expression [22]. Caveolin-1 (Cav1), a scaffold protein, is the most well-known among the main components of caveolae [23]. Cav1 is involved in signaling transduction, cell migration, cell proliferation, cell polarity, cholesterol, and cellular homeostasis [24]. Cav1 recruits β-catenin and E-cadherin to caveolae membrane domains further to stabilize the cell–cell adhesion junctions in epithelial cells [25]. Numerous oncogenes, such as Src, Ras, and Bcr-Abl, have been discovered to downregulate Cav1 expression [26]. Moreover, Cav1 was also downregulated or absent in Ha-Ras(-) and Src(-) transgenic mice [27]. The restoration of Cav1 suppressed cell proliferation in benign tumor cells, and is a tumor suppressor in the early stage of cancer [28].

In this study, we sought to investigate whether the downregulation of Cav1 induced by Ha-Ras^V12^ overexpression is relevant to actin belt disruption and YAP nuclear translocation to trigger multilayer cellular aggregate formation. As expected, Ha-Ras^V12^ suppressed Cav1 expression to disrupt cell junction integrity and promote YAP nuclear translocation. Surprisingly, nuclear YAP induced by Ha-Ras^V12^ overexpression is not required for multilayer cellular aggregate formation. Our data revealed that the activation of Rac is required and sufficient for multilayer cellular aggregate formation.

## 2. Materials and Methods

### 2.1. Cells and Culture Conditions

Madin–Darby Canine Kidney (MDCK) cells and MK4 cells (MDCK-transfected cells harboring pSVlacORas and pHblacINLSneo plasmids) [22] were cultured in Dulbecco’s Modified Eagle’s Medium (DMEM, Gibco, Carlsbad, CA, USA) supplemented with 5% calf serum (HyClone, Logan, UT, USA) and Penicillin–Streptomycin (Sigma-Aldrich, St. Louis, MO, USA). All cell lines were maintained at 37 °C in a 5% CO_2_ humidified incubator.

### 2.2. Plasmids, shRNA, siRNA, and Transfection

The Caveolin-1-Myc-mRFP plasmid was kindly gifted by Dr. Ivan R Nabi [29]. The short hairpin RNA (shRNA) constructs, such as shLacZ, shCav1 #1, and shCav1 #2, were generated as previously described [22]. Constitutively active and dominant-negative Rho-family GTPase (Rac1) were gifted by Dr. Bing-Cheng Wang [30]. The transfection of the plasmid was performed as previously described [22]. To generate clones expressing stable Cav1, flow cytometry was used to sort the cells to enrich the mRFP-positive cells. Geneticin (G418, Sigma, St. Louis, MO, USA) was also used to select clones expressing stable plasmids. Lentiviral shRNA vectors were used to silence genes as previously described [22]. Cells were selected by using puromycin (Sigma).

### 2.3. RT-PCR

TRIzol reagent (Life Technologies, Carlsbad, CA, USA) was used to extract the total RNA from cells following the manufacturer’s instructions. The RNA quality was verified by NanoDrop (Thermo Scientific, Waltham, MA, USA) and reverse-transcribed by using the RevertAid First Strand cDNA Synthesis Kit according to the manufacturer’s protocol (Thermo Scientific). cDNA was used as a template for PCR using specific primers for the following genes: dog CTGF (forward: 5′-GAG GAA CTG TGT ACC GGA GC-3′; reverse: 5′-AAC AGG CAC TCC ACT CTG TG-3′); dog CYR61 (forward: 5′-CCC AGT TTT TGG ACG GAG C-3′; reverse: 5′-CAT TTC TTG CCC TTC AGG CT-3′) and β-actin (forward: 5′-ACT GGG ACG ACA TGG AGA AG-3′; and reverse: 5′-GGT ACG ACC AGA GGC ATA CAG-3′). PCR was performed as previously described [22].

### 2.4. Western Blot Analyses

Cells were harvested by using a lysis buffer (20 mM Tris-HCl, pH 8, 150 mM NaCl, 0.1% Triton X-100, 2 mM EDTA) containing 1 × protease inhibitor cocktail (complete). Freshly prepared phosphatase inhibitor (Halt), 1 mM Na₃VO₄, 1 mM PMSF, and 1 mM NaF were added before use. Western blot analysis was performed as previously described [31]. Primary antibodies were used against the following proteins: from BD Biosciences Pharmingen; YAP and Cav1 from Abcam (Cambridge, MA, USA); from Invitrogen; pYAP, Lamin a/c, LATS1, MOB, MST1, MST2, SAV1, pJNK, pERK, and ERK from Cell Signaling (Boston, MA, USA); Pan-Ras from Calbiochem; JNK from Santa Cruz Biotechnology, Inc. (Santa Cruz, CA, USA); myosin IIA and myosin IIB from Sigma-Aldrich; and GAPDH from GeneTex (Irvine, CA, USA).

### 2.5. Cell Fractionation

To obtain protein from different cell fractions, we used the REAP (rapid, efficient, and practical) method, which is a 2 min cell fractionation method [32]. Cells (2 × 10⁶) were seeded in a 10 cm dish in an overconfluent experiment. The cells were rinsed with PBS and collected in 1 mL of PBS. After centrifuging at 12,000 rpm for 10 s, the supernatant was removed. Cell pellets were suspended in 0.1% NP40. Lysate (300 µL) was collected as “whole cell lysate”, and further lysed using lysis buffer. The remaining lysate (700 µL) was centrifuged at 12,000 rpm for 10 s. Then, the supernatants of the lysate as the “cytosolic fraction” and cell pellet of the lysate as the “nuclear fraction” were treated with lysis buffer in order to quantify the cell concentrations for further Western immunoblot analysis.

### 2.6. Pull-Down Assay

To obtain the Rac1 activity, we used the Rac1 Activation Assay Kit (Merck Millipore, MA, USA). For the overconfluent experiment, 2 × 10⁶ cells were plated in a 10 cm dish. After rinsing the cells with PBS, the cells were harvested with 1 mL ice-cold Mg^2^⁺ lysis/wash buffer (MLB) (containing 10% glycerol, 10 µL/mL leupeptin, and 10 µL/mL aprotinin). After quantifying the cell lysate concentration, 10 µL of Rac assay reagent (PAK-1 PBD, agarose) was applied to 2.5 mg of the protein sample. Then, the reaction mixture was gently agitated for 1 h at 4 °C. Next, the agarose beads were collected by centrifuging at 14,000× *g* for 10 s. After washing the beads three times with MLB buffer, the sample underwent Western immunoblot analysis.

### 2.7. Immunofluorescence Staining and Confocal Microscopy

Immunofluorescence staining was performed as previously described [31]. The following primary antibodies were used: myosin IIA and myosin IIB (Sigma-Aldrich); Cav1 (BD Biosciences, Franklin Lakes, NJ, USA) (Pharmingen; San Jose, CA, USA); and YAP and Cav1 (Abcam) (Cambridge, MA, USA). After the cells were washed with PBS, they were incubated with anti-rabbit or anti-mouse IgG antibodies conjugated with Alexa 488 or 594 (Invitrogen-Molecular Probes, Eugene, OR, USA) and/or Alexa 594 or 647 Phalloidin and 10 µg/mL Hoechst 33,258 (Sigma-Aldrich) for 1 h at room temperature. Imaging was performed from sequential z-series scans using an FV1000-IX81-HSD confocal microscope, FV1000-BX61WI (multiphoton laser scanning microscope), or confocal laser scanning microscope FV3000-IX83 (Olympus, Tokyo, Japan) with a 60× oil immersion lens.

### 2.8. Crystal Violet Staining

Cells were washed with PBS and fixed with 4% paraformaldehyde in PBS for 1 h at room temperature. The paraformaldehyde was washed away with PBS and the cells were stained with 0.025% crystal violet for 10 min. The cells were washed with PBS until they were clear. Phase-contrast images were captured by inverted microscopy (Leica DM IRB) with a 10× lens. The area of multilayer cellular aggregates was analyzed using ImageJ software.

### 2.9. Anchorage-Independent Growth Assay

The ability of anchorage-independent growth was determined by the number of colony formations in the soft agar system. The soft agar system contained 1.5 mL of base agar (0.5% low-melting agarose, 1× high-glucose DMEM, and 5% CCS) suspended with 5 × 10^3^ cells in 1.5 mL of top agarose (0.35% low-melting agarose, 1× high-glucose DMEM, and 5% CCS) in a 35 mm dish. After solidification, 2 mL of high-glucose DMEM supplemented with 5% CCS was added to the soft agar, then incubated under a 5% CO₂ atmosphere at 37 °C. The culture medium was replenished every three days. After 14 days, the sample was fixed with 4% paraformaldehyde overnight. Then, the sample was stained with 0.025% crystal violet for a further 1 h. The soft agar was washed with PBS until it became clear. Images were taken using a camera (Alphaimager 2200, Alpha Innotech Co., San Leandro, CA, USA).

### 2.10. Statistical Analysis

Data are expressed as the mean ± S.E.M. and are representative of experiments performed at least three times independently. The quantification results of YAP nuclear localization were defined by the percentage of cells with predominant nuclear YAP localization (N > C, >110%), equal nuclear and cytosol YAP localization (N = C, 90–110%), and predominant cytosol YAP localization (N < C, <90%). The definition of YAP was based on the expression intensity of YAP in the nucleus divided by the expression intensity of YAP in the cytosol. We set a small range (0.9–1.1) as the equal nuclear and cytosol YAP localization. Values outside this range were either classified as predominant nuclear YAP localization or predominant cytosol YAP localization. GraphPad Prism 8.0 (GraphPad Software, San Diego, CA, USA) was used to calculate the plot mean and standard error of the mean (SEM) of the measured quantities, and significances were assessed by one-way ANOVA. The results were analyzed via ANOVA and *t*-tests. Statistical significance was determined with *p*-values < 0.05.

## 3. Results

### 3.1. Ha-Ras^V12^ Facilitates YAP Nuclear Translocation through Downregulated Junctional Myosin IIB

Our previous studies demonstrated that Ha-Ras^V12^ overexpression results in the cell softening and transformation of MK4 cells (MDCK harboring inducible Ha-Ras^V12^ gene) [33]. After reaching confluence (>2 days after cell plating), MK4 cells displayed cytoplasmic YAP and well-organized actin belt formation. Upon IPTG induction, the overexpression of Ha-Ras^V12^ induced YAP nuclear translocation and disrupted the actin belt integrity (Figure 1a–c). Western blot analysis of nuclear/cytoplasmic fractionations confirmed that Ha-Ras^V12^ overexpression caused the dephosphorylation and nuclear translocation of YAP (Appendix A). Consequently, the expression of YAP-TEAD downstream genes, such as CTGF, was upregulated in Ha-Ras^V12^-overexpressed cells (Figure 1d,e). However, Hippo pathway-associated components were not downregulated after Ha-Ras^V12^ induction (Appendix A), suggesting that the Hippo pathway is irrelevant to Ha-Ras^V12^-induced YAP activation in MK4 cells at the overconfluent stage. Non-muscle myosin II is a well-known component of the actin belt, which is a dense circumferential belt of actin filaments located on the cytoplasmic face of the adherens junction [7]. Furukawa et al. demonstrated that the global inhibition of myosin with blebbistatin disrupted the actomyosin tension of the actin belt and led to YAP nuclear translocation in high-density cells [8]. To understand whether myosin II is involved in Ha-Ras^V12^-induced YAP nuclear translocation in MDCK cells at the confluent stage, we assessed the expression and distribution of myosin IIA and myosin IIB. In normal MK4 cells, myosin IIA was present in the cytosol and myosin IIB was lined with the junctional actin and located at the adherens junction (Figure 1c and Appendix A). The induction of Ha-Ras^V12^ did not change the protein levels of myosin IIA and myosin IIB (Figure 1f,g) and the distribution of myosin IIA (Appendix A). Meanwhile, the junctional actin belts were disrupted and accompanied by dislocated myosin IIB (Figure 1c). Myosin IIB dislocated from the cell junction to the cytosol after Ha-Ras^V12^ induction.

To further confirm the role of myosin IIB-related actomyosin tension in regulating YAP nuclear translocation, we used myosin II inhibitors, including ML7 (myosin II light-chain kinase inhibitor) and blebbistatin (myosin II heavy-chain inhibitor). As shown in Figure 2a,b, the treatment with blebbistatin did not destroy the junctional actin structure in MK4 cells. Both ML7 and blebbistatin caused the redistribution of myosin IIB from the junction to the cytosol (Figure 2a) and enhanced nuclear YAP translocation (Figure 2b,c). Taken together, these data revealed that the overexpression of Ha-Ras^V12^ suppressed junctional myosin IIB, which might be relevant for YAP nuclear translocation.

### 3.2. Cav1 Preserved Junctional Myosin IIB and Restricted Ha-Ras^V12^-Induced YAP Nuclear Translocation

Caveolin-1 plays an essential role in maintaining cell junction integrity by recruiting the E-cadherin–catenin complex to caveolae [25]. Our previous study showed that Cav1 was downregulated upon Ha-Ras^V12^ induction. Moreover, Cav1 abolishes not only Ha-Ras^V12^-induced cell scattering, but also the disorganization of junctional protein [22]. In this study, we first knocked down Cav1 with specific shRNA in both MDCK and MK4 cells. The Western blot results showed the knockdown efficiency of Cav1 in MK4 and MDCK cells (Appendix A). Specifically, the results showed that the knockdown efficiency of Cav1 was higher in MK4 cells compared to that in MDCK cells. The depletion of Cav1 decreased the junctional myosin IIB expression and disturbed the actin belt integrity in MK4 cells (Figure 3a). In contrast, the downregulation of Cav1 in MDCK only resulted in slightly junctional myosin IIB downregulation and actin belt disruption (Appendix A). Meanwhile, nuclear YAP retention became prominent at the confluent stage in both MDCK and MK4 cells (Figure 3b,c and Appendix A). Collectively, these data suggest that Cav1 is required for stabilizing junctional myosin IIB and actin belt integrity, so as to suppress YAP nuclear translocation.

To confirm the role of Cav1 in modulating the junctional myosin IIB distribution and YAP nuclear translocation, we further overexpressed Cav1-conjugated RFP in MK4 cells. We found that Cav1 overexpression prevented Ha-Ras^V12^-induced junctional myosin IIB downregulation. The overexpression of Cav1 in MK4 cells also displayed a well-organized actin belt structure, regardless of Ha-Ras^V12^ induction (Figure 3d). Furthermore, Cav1 overexpression abolished Ha-Ras^V12^-induced YAP nuclear translocation at the overconfluent stage (Figure 3e,f).

### 3.3. Ha-Ras^V12^ Downregulated Cav1 Expression and YAP Nuclear Retention through MEK Pathway

We already know that Ha-Ras^V12^ overexpression downregulates the Cav1 expression and changes the Cav1 distribution (22). To investigate the mechanism of Cav1 downregulation triggered by Ha-Ras^V12^ overexpression, we evaluated the ERK expression, which is a downstream signal of Ras. The results showed that the overexpression of Ha-Ras^V12^ induced ERK activation (Figure 4b). Moreover, the inhibition of ERK activation using MEK inhibitor U0126 prevented the downregulation of Cav1 and YAP nuclear translocation in Ha-Ras^V12^-overexpressed cells (Figure 4). Some studies have shown that the activation of JNK and Rac1—downstream signals of MAPK—is relevant to YAP nuclear retention [34,35]. We next wanted to assess whether the activation of JNK and Rac1 is required for Ha-Ras^V12^-induced YAP nuclear retention. Cells were treated with SP600125 (JNK inhibitor) and EHT1864 (Rac1 inhibitor). Both SP600125 and EHT1864 failed to prevent Ha-Ras^V12^-induced YAP nuclear translocation (Figure 4d,e). In summary, the results confirmed that Ha-Ras^V12^ downregulated Cav1 expression through ERK activation and further induced YAP nuclear translocation.

### 3.4. Nuclear YAP Is Not Involved in Ha-Ras^V12^-Induced Cellular Aggregation

After determining the regulation of YAP nuclear translocation in Ha-Ras^V12^-overexpressing cells, we sought to investigate the function of Ha-Ras^V12^-induced nuclear YAP. To assess the function of nuclear YAP, we employed verteporfin (VP, YAP inhibitor) to suppress YAP nuclear translocation in Ha-Ras^V12^-overexpressing cells. VP successfully abolished Ha-Ras^V12^-induced YAP activation at the overconfluent stage (Figure 5a,b). VP also downregulated Ha-Ras^V12^-induced YAP-TEAD downstream genes CYR61 mRNA expression (Figure 5c,d). Nuclear YAP has been reported to be involved in cellular aggregation in MDCK cells at the confluent stage [14]. In our study, we also found that the overexpression of Ha-Ras^V12^ triggered multilayer cellular aggregation at the overconfluent stage (Figure 5e,f). We propose that nuclear YAP induced by Ha-Ras^V12^ overexpression may require multilayer cellular aggregation. Surprisingly, the inhibition of nuclear YAP did not suppress Ha-Ras^V12^-induced multilayer cellular aggregation (Figure 5e,f). Interestingly, one study determined that YAP regulates anchorage-independent growth in prostate cancer cells [36]. We indicated that the overexpression of Ha-Ras^V12^ induced colony formation, which can be suppressed by VP (Figure 5g,h). This result suggests that nuclear YAP is required for colony formation, instead of multilayer cellular aggregation.

### 3.5. Rac1 Activity Is Required for Ha-Ras^V12^-Induced Cellular Aggregation

In other studies, YAP has been found to be necessary for cellular aggregation in heterogeneous cultures [14]. YAP-activated cells were co-cultured with normal MDCK cells to observe cellular aggregation formation. In our study, we found that the overexpression of Ha-Ras^V12^ induced multilayer cellular aggregation in both homogeneous and heterogeneous cells (Figure 6a and Appendix A). Ha-Ras^V12^ disrupted the cell junction integrity and disturbed actin filament organization. We assumed that cytoskeleton reorganization may be involved in Ha-Ras^V12^-induced cellular aggregation. Our previous study revealed that Ha-Ras^V12^ promotes RhoA activity, which is relevant for cytoskeleton organization [33]. RhoA activates Rho-associated protein kinase (ROCK), which is relevant for myosin light-chain (MLC) activation. The activation of MLC facilitated actin reorganization. Therefore, we applied cells with Y27632 (ROCK inhibitor) to indicate whether the inhibition of ROCK prevented Ha-Ras^V12^-induced multilayer cellular aggregate formation. Surprisingly, the inhibition of ROCK triggered multilayer cellular aggregates in MK4 cells without Ha-Ras^V12^ induction (Figure 6a,b). Some studies have stated that the inhibition of Rho-associated protein kinase (ROCK) facilitated cell extrusion in MDCK cells [18,37]. Moreover, the inhibition of ROCK has also been reported to induce Rac1 activity [38]. We also found that both the inhibition of ROCK and induction of Ha-Ras^V12^ upregulated Rac activity (Figure 6c,d). To determine whether Rac activity is required for multilayer cellular aggregate formation, we applied cells with combined treatment, including Y27632 and EHT1864 (Rac inhibitor). It was demonstrated that EHT1864 prevented both Ha-Ras^V12^- and Y27632-induced multilayer cellular aggregates. Furthermore, to confirm whether the activation of Rac is sufficient for multilayer cellular aggregation, we transfected different forms of Rac-conjugated GFP in MDCK and MK4 cells, including Rac1-wild type (Rac1-WT), Rac1 constitutive active form (Rac1-L), and dominant-negative form (Rac1-N). The results demonstrated that only cells overexpressing the Rac constitutive active form (Rac1-L) triggered cellular aggregation at the confluent stage (Figure 6e,f and Appendix A). Collectively, the activation of Rac1 is required and sufficient for cellular aggregation at the confluent stage.

### 3.6. Ha-Ras^V12^-Induced Multilayer Cellular Aggregates through ERK-Rac Pathway

We further sought to investigate the mechanism of Ha-Ras^V12^-induced Rac activity. The ERK pathway has been reported to activate Rac activity in BRAF- and NRAS-mutant melanoma [39]. To evaluate whether ERK is relevant to Ha-Ras^V12^-induced Rac activation and multilayer cellular aggregate formation, the cells were treated with U0126. It was found that Ha-Ras^V12^ triggered Rac activation and cellular aggregate formation, which was suppressed by U0126 (Figure 7). This suggests that Ha-Ras^V12^ induced ERK activation and further upregulated Rac activity to trigger multilayer cellular aggregate formation.

## 4. Discussion

In this study, we demonstrated the mechanism underlying Ha-Ras^V12^-induced YAP nuclear translocation and multilayer cellular aggregates. We found that Ha-Ras^V12^ overexpression induced YAP nuclear translocation through the disturbance of the cell junction and dislocated myosin IIB. Our previous study revealed that Ha-Ras^V12^ decreased Cav1 expression [22]. Our data showed that the overexpression of Cav1 abolished Ha-Ras^V12^-induced myosin IIB delocalization and YAP nuclear translocation in MK4 cells. Moreover, the knockdown of Cav1 resulted in the downregulation of junctional myosin IIB and YAP nuclear retention in MDCK and MK4 cells. U0126 prevented Ha-Ras^V12^-induced Cav1 downregulation. This suggests that Ha-Ras^V12^ induced Cav1 downregulation and YAP nuclear translocation through the MEK signaling pathway. Nevertheless, the activation of YAP was required for Ha-Ras^V12^-induced independent growth, but not multilayer cellular aggregates. Next, we demonstrated Ha-Ras^V12^-induced Rac1 activation through the MAPK signaling pathway, resulting in multilayer cellular aggregates in MK4 cells (Figure 8).

The role of YAP in multilayer cellular aggregate formation remains controversial. Bin Zhao et al. demonstrated that the overexpression of YAP-WT constitutively formed (YAP-5SA) in MDCK cells triggered multilayer cellular aggregates when YAP-transformed cells were surrounded by normal cells, whereas the overexpression of YAP-S94A—which eliminated TEAD binding—prevented multilayer cellular aggregate formation [14]. In our study, we observed multilayer cellular aggregate formation in the entire population of Ha-Ras^V12^-overexpressed cells. The overexpression of Ha-Ras^V12^ facilitated cell transformation, which resulted in YAP nuclear translocation in all bottom fractions of cells; however, YAP was located in the cytosol inside the cellular aggregate area (Appendix A). However, the reason for YAP being located in the cytosol in the cellular aggregate area remains unknown and requires further investigation. Additionally, only a handful of studies support the idea of nuclear YAP not being required for Ha-Ras^V12^-induced multilayer cellular aggregate formation. Firstly, VP disturbed YAP/TEAD binding and prevented Ha-Ras^V12^-induced YAP nuclear retention; however, it failed to abolish Ha-Ras^V12^-induced multilayer cellular aggregate formation (Figure 5e,f). Secondly, we confirmed that Rac is required for cellular aggregate formation. Lastly, the inhibition of Rac failed to prevent Ha-Ras^V12^-induced YAP nuclear translocation.

The activation of YAP can be triggered by diverse mechanotransductions, such as low cell density, stiff ECM, high cell-spreading area, and cell stretching [40]. The underlying mechanism for the regulation of YAP nuclear translocation may differ from various stimulations. Cell junction integrity and actomyosin tension have been shown to regulate YAP nuclear translocation [41,42]. We demonstrated that Cav1 overexpression restored Ha-Ras^V12^-induced cell junction disruption and YAP nuclear translocation. Furthermore, the knockdown of Cav1 and inhibition of myosin II disturbed junctional myosin IIB and subsequently caused YAP nuclear translocation in MK4 cells at the overconfluent stage. Therefore, understanding how Cav1 modulates cell junction integrity and myosin IIB distribution will be an important area for future research. In our study, we found that U0126 prevented Ha-Ras^V12^-induced Cav1 downregulation. Cav1 has also been found to play an inhibitory role in the Ras/MAPK cascade [43]. Furthermore, the regulation of the Ras/MAPK cascade is complicated, and the overexpression of Cav1 may prevent the activation of the Ras/MAPK cascade. Considering the lower Cav1 expression in several cancers, these data highlight the role of tumor suppressor Cav1 in the initiation of tumorigenesis.

Actomyosin dynamics are essential for adherens junction formation and functioning during epithelial tissue homeostasis [44,45]. Rho GTPases play an important role in regulating junctional actin and myosin [46]. Rho-associated coiled-coil kinase (ROCK) activates myosin II contractility through the phosphorylated myosin light chain. Rac is a member of the Rho GTPases family. The activation of Rac1 in the initiation of cell–cell contact facilitates cadherin ligation [20]. After cell junction maturation, actin filaments are organized with myosin II into bundles, leading to a functional contractile structure [47]. Moderated Rac1 activity functions to stabilize cadherin-dependent adhesion [21]. However, the overactivation of Rac1 in Ha-Ras^V12^-transformed cells results in the disassembly of cadherin contacts. Whether and how Rac1 activity is required for Ha-Ras^V12^-induced cellular and mechanical transformation remains to be further elucidated.

## Figures and Tables

**Figure 1 biomedicines-10-00977-f001:**
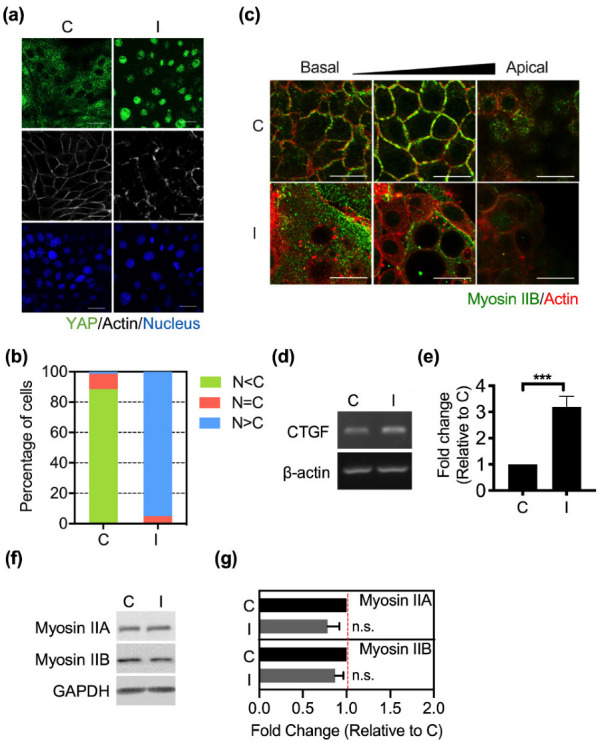
Ha-Ras^V12^ disrupted the cell junction integrity and induced YAP nuclear translocation at the overconfluent stage: (**a**) MK4 cells at the overconfluent stage were treated with (I) or without (C) IPTG (5 mM) for 24 h. Representative confocal images of cells, which were stained with YAP (green), Hoechst 33,258 (blue), and Phalloidin 647 (gray). Scale bar = 20 µm; (**b**) Percentage of cells with predominant nuclear YAP localization (N > C), equal nuclear and cytosol YAP localization (N = C), and predominant cytosol YAP localization (N < C); (**c**) MK4 cells were treated with or without IPTG (5 mM) for 24 h. Representative confocal images of cells, which were stained with myosin IIB (green) and Phalloidin (red). The arrow indicates the different z sections of cells from the basal to the apical level. Scale bar = 20 µm; (**d**,**e**) Representative PCR and quantification results of the YAP downstream gene expression of CTGF; (**f,g**) Representative immunoblots and quantification results of tensional proteins, including myosin IIA and myosin IIB. All data are expressed as the mean ± SEM from three independent experiments. n.s.: Not significant, *** *p* < 0.001.

**Figure 2 biomedicines-10-00977-f002:**
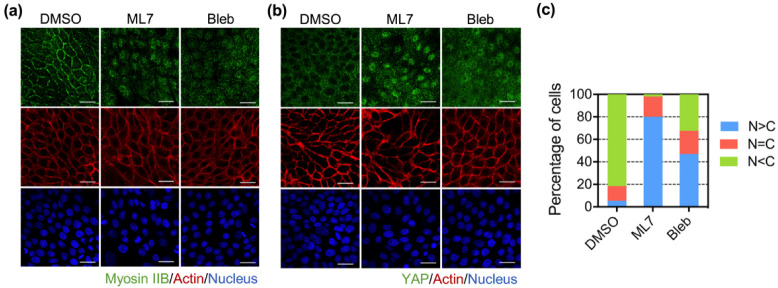
Inhibition of junctional myosin IIB promoted YAP nuclear retention. MK4 cells at the overconfluent stage were treated with the indicated inhibitors for 24 h. The inhibitors included ML7 (25 µM, myosin light-chain kinase inhibitor) and blebbistatin (Bleb, 50 µM, myosin II heavy-chain inhibitor), dimethyl sulfoxide (DMSO) was used as control: (**a**) Representative confocal immunofluorescence images of MK4 cells at the confluent stage, which were stained with myosin IIB (green), Phalloidin (red), and Hoechst 33,258 (blue). Scale bar = 20 µm; (**b**) Representative confocal immunofluorescence images of MK4 cells at the confluent stage, which were stained with YAP (green), Phalloidin (red), and Hoechst 33,258 (blue). Scale bar = 20 µm; (**c**) Percentage of cells with predominant nuclear YAP localization (N > C), equal nuclear and cytosol YAP localization (N = C), and predominant cytosol YAP localization (N < C).

**Figure 3 biomedicines-10-00977-f003:**
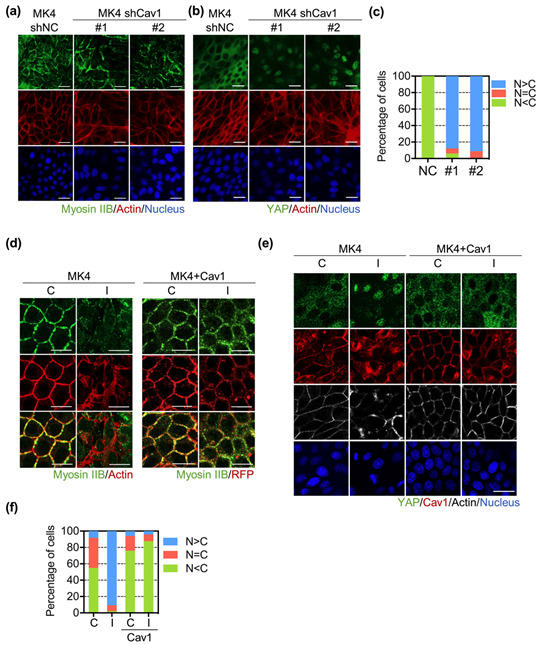
Downregulation of Cav1 mediated Ha-Ras^V12^-induced cell junction disruption and YAP nuclear translocation. Knockdown of Cav1 in MK4 cells cultured at overconfluent stage. Two MK4 shCav1 cell lines were used, including MK4 shCav1 #1(#1) and MK4 shCav1 #2 (#2). MK4 shNC (NC) was used as knock-down control cell line. (**a**) Representative immunofluorescence images of MK4 cells at the confluent stage, which were stained with myosin IIB (green), Phalloidin (red), and Hoechst 33,258 (blue). Scale bar = 20 µm. (**b**) Representative immunofluorescence images of MK4 cells at the confluent stage, which were stained with YAP (green), Phalloidin (red), and Hoechst 33,258 (blue). Scale bar = 20 µm. (**c**) Percentage of cells with predominant nuclear YAP localization (N > C), equal nuclear and cytosol YAP localization (N = C), and predominant cytosol YAP localization (N < C). MK4 cells were overexpressed with RFP-conjugated Cav1. Cells’ overexpression of Cav1 was treated with or without IPTG (5 mM) for 24 h at the overconfluent stage. (**d**) Representative confocal images of MK4 cells at the confluent stage, which were stained with myosin IIB (green) and Phalloidin (red). Scale bar = 20 µm. (**e**) Representative confocal images of MK4 cells at the confluent stage. Cells were stained with YAP (green), Cav1 (red), Phalloidin (gray), and Hoechst 33,258 (blue). Scale bar = 20 µm. (**f**) Percentage of cells with predominant nuclear YAP localization (N > C), equal nuclear and cytosol YAP localization (N = C), and predominant cytosol YAP localization (N < C).

**Figure 4 biomedicines-10-00977-f004:**
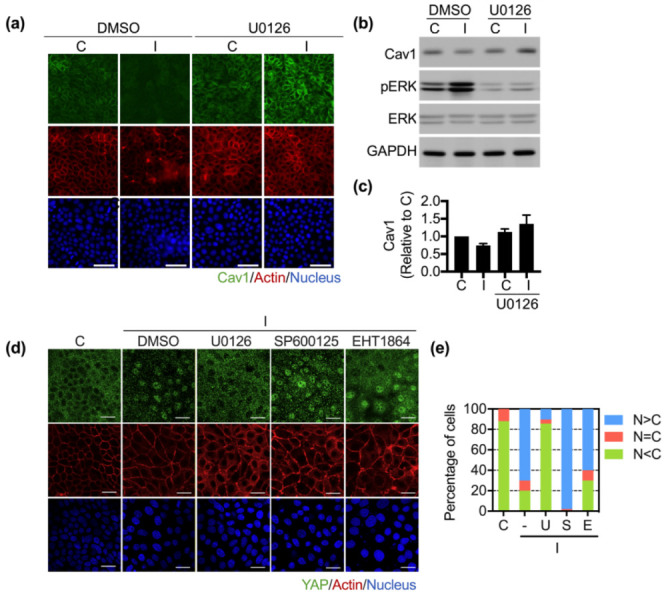
MEK is required for Ha-Ras^V12^-induced YAP nuclear translocation and Cav1 downregulation. MK4 cells at the overconfluent stage were pre-treated with or without U0126 (20 µM, MEK inhibitor) for 30 min, then treated with or without IPTG (5 mM) for a further 24 h: (**a**) Representative immunofluorescence images of MK4 cells at the confluent stage, which were stained with Cav1 (green), Phalloidin (red), and Hoechst 33,258 (blue). Scale bar = 50 µm. (**b**,**c**) Representative immunoblots and quantification results of Cav1, pERK, and ERK. MK4 cells at the overconfluent stage were pretreated with the indicated inhibitors for 30 min before administering with or without IPTG (5 mM) for a further 24 h. The inhibitors included U0126 (U, 20 µM), SP600125 (S, 10 µM, JNK inhibitor), or EHT1864 (E, 10 µM, Rac1 inhibitor). (**d**) Representative confocal images of MK4 cells at the confluent stage, which were stained with YAP (green), Phalloidin (red), and Hoechst 33,258 (blue). Scale bar = 20 µm. (**e**) Percentage of cells with predominant nuclear YAP localization (N > C), equal nuclear and cytosol YAP localization (N = C), and predominant cytosol YAP localization (N < C). All data are expressed as the mean ± SEM from three independent experiments.

**Figure 5 biomedicines-10-00977-f005:**
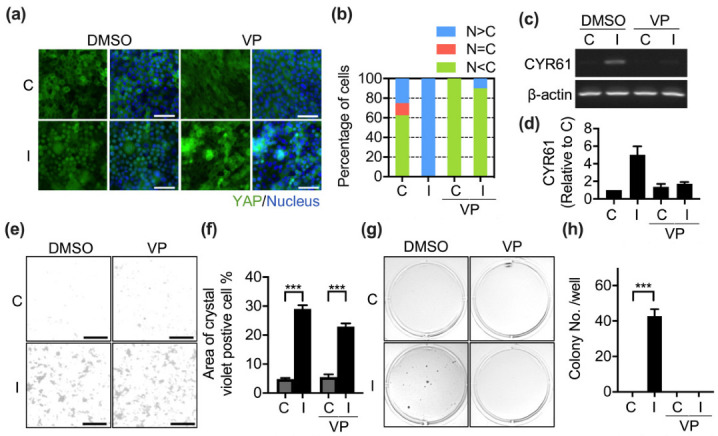
Activation of YAP is required for Ha-Ras^V12^-induced colony formation, but not multilayer cellular aggregation. MK4 cells at the overconfluent stage were pretreated with verteporfin (VP, 5 µM, YAP-TEAD binding inhibitor) for 2 h before administering with or without IPTG (5 mM) for a further 24 h: (**a**) Representative immunofluorescence images of cells at overconfluent stage, which were stained with YAP (green) and Hoechst 33,258 (blue). Scale bar = 50 µm. (**b**) Percentage of cells with predominant nuclear YAP localization (N > C), equal nuclear and cytosol YAP localization (N = C), and predominant cytosol YAP localization (N < C). (**c**,**d**) Representative RT-PCR and quantification results of CYR61, a YAP downstream target gene. β1-actin served as the internal control. (**e**) Representative crystal violet-staining images of cells treated with or without the indicated conditions. Scale bar = 200 µm. (**f**) Quantification results of the crystal violet-positive cell area from (**e**). (**g**) Anchorage-independent growth (colony formation assay) of cells treated for the indicated conditions in soft agar for 14 days. (**h**) Quantification results of the soft agar assay from (**g**). All data are expressed as the mean ± SEM from three independent experiments. *** *p* < 0.001.

**Figure 6 biomedicines-10-00977-f006:**
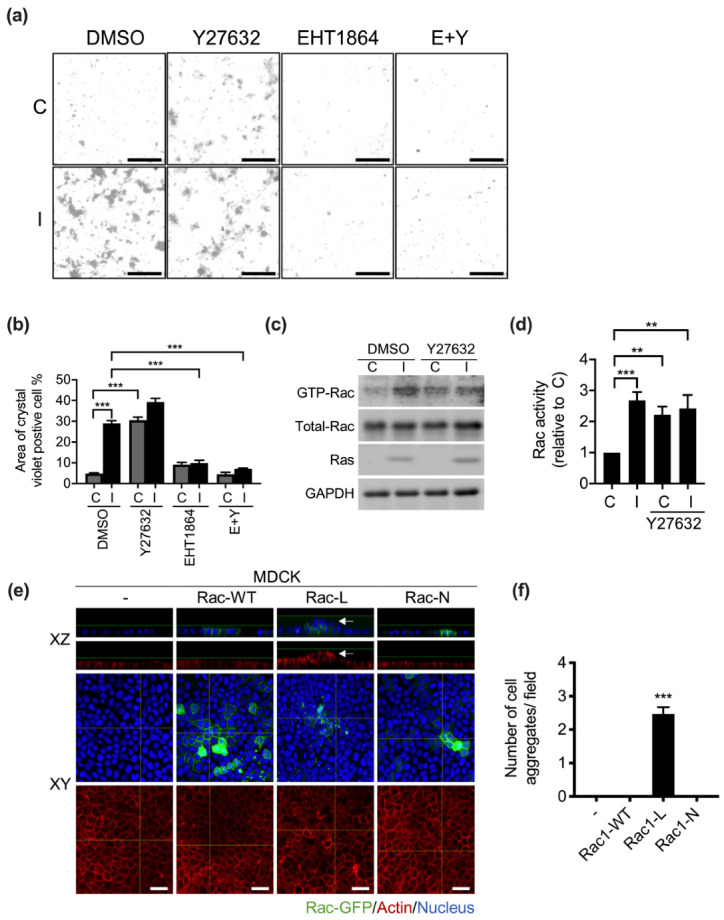
Activation of Rac1 is necessary and sufficient for multilayer cellular aggregates. MK4 cells at the overconfluent stage were pretreated with indicated inhibitors for 2 h before administering with or without IPTG (5 mM) for a further 24 h. Inhibitors included Y27632 (Y, 10 µM, ROCK inhibitors) or EHT1864 (E, 10 µM, Rac1 inhibitor): (**a**) Representative crystal violet-staining images of cells treated with or without indicated conditions. Scale bar = 200 µm. (**b**) Quantification results of crystal violet-positive cell area from (**a**). (**c**) Representative immunoblot results for GTP-Rac, Rac, and Ras. GAPDH served as the internal control. GTP-Rac was extracted by a pull-down assay as described in the Materials and Methods. (**d**) Quantification results of Rac activity from C. MDCK cells were transfected with GFP-conjugated Rac1-WT (Rac1-wild type), Rac1-L (Rac1 constitutive active form), and Rac1-N (dominant-negative form) transiently. (**e**) Representative confocal xy projections with z stack and xz cross-section images showing Rac1-GFP (green), and the cells were stained with Phalloidin (red) and Hoechst 33,258 (blue). White arrowheads indicate multilayer cellular aggregates. Scale bar = 20 µm; (**f**) Quantification results for the number of cellular aggregates per field in 200 × 200 µm^2^ from (**e**). All data are expressed as the mean ± SEM from three independent experiments. ** *p* < 0.01, *** *p* < 0.001.

**Figure 7 biomedicines-10-00977-f007:**
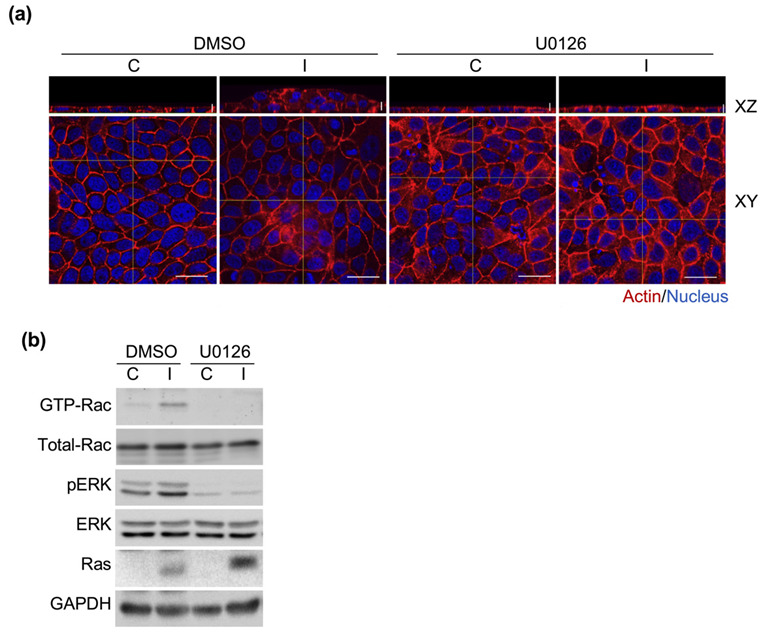
Ha-Ras^V12^ triggered Rac activity through the MEK pathway. MK4 cells were treated with or without U0126 (20 µM) before administering with or without IPTG (5 mM) for a further 24 h: (**a**) Representative confocal images and xz cross-section images of MK4 cells at the confluent stage, which were stained with Phalloidin (red) and Hoechst 33,258 (blue). Scale bar = 20 µm; (**b**) Results of representative immunoblots for GTP-Rac, Rac, pERK, ERK, and Ras. GAPDH served as the internal control.

**Figure 8 biomedicines-10-00977-f008:**
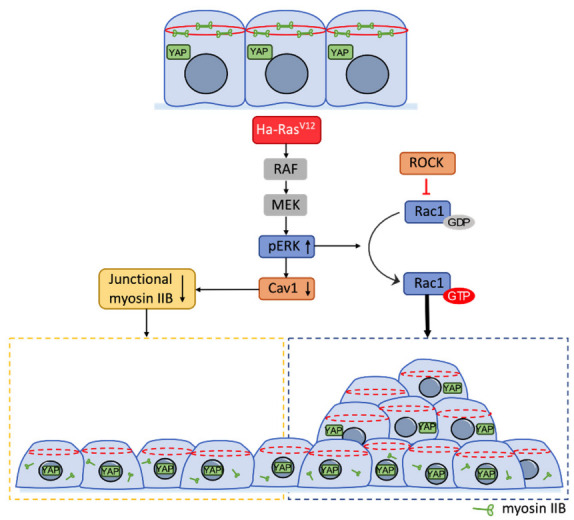
Mechanism of Ha-Ras^V12^-induced YAP nuclear translocation and multilayer cellular aggregates. Ha-Ras^V12^ downregulated Cav1 through the ERK pathway, and further suppressed junctional myosin IIB so as to promote YAP nuclear retention. Ha-Ras^V12^ also induced the ERK-Rac pathway to trigger multilayer cellular aggregate formation at the overconfluent stage. ↑ indicates upregulation and ↓ indicates downregulation.

## Data Availability

The data that support the findings of this study are available from the corresponding author upon reasonable request.

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
