# Peer review of "Ha-RasV12-Induced Multilayer Cellular Aggregates Is Mediated by Rac1 Activation Rather Than YAP Activation"

_biomedicines, 2022, doi:10.3390/biomedicines10050977_

Round 1

Reviewer 1 Report

The authors show that Rac-V12 mediated multicellular aggregates is dependent on Rac1 activation and is independent of YAP1 activity. The authors convincingly show that Rac1 activation is sufficient for multilayer cellular aggregate formation and that restricting YAP1 do not inhibit the process. Overall, the study is good but observational. It lacks a mechanism on several grounds:

  1. What is the role of Myosin IIB in the overall process?
  2. Is Myosin IIA function redundant for Myosin IIB?
  3. How is Rac1 activated? What is the specific GEF that functions here?
  4. How do Rac1 specifically induce the effect?

Some specific comments:

  1. In Fig 1c we can hardly see any Myo IIB translocation. The authors need to provide better images and or clarify how they quantify from these images.
  2. Define the stress fiber reorganization after Myosin IIB translocation to the cytosol. How do the transverse arcs, dorsal SF and Ventral SF look? Set up rescue experiments with Myo IIB and Myo IIA to attribute specific roles for them.
  3. Overall, the quality of the images has to improve to visualize the conclusions described in the text.
  4. How do Rac1 activity influence the cortical actin structure to form aggregates?

Reviewer 2 Report

Dear Authors,

Thank you for submitting the manuscript titled, “Ha-RasV12-induced multilayer cellular aggregates is mediated 2 by Rac1 activation rather than YAP activation” written by Wu, et al. The data looks nice, however, too many grammatical errors and typos are seen in the text and I can not decide if this manuscript is suitable to the journal. Please correct the errors and ask a native English speaker/writer to do proofreading.

Thank you for your consideration.

Round 2

Reviewer 1 Report

The authors have satisfactorily answered to my queries with sufficient explanations and/or improved experiments. I would request the manuscript to be accepted in the current form

Author Response

Thanks for the positive comment.

Reviewer 2 Report

Dear Authors,

Thank you very much for submitting the article titled “Ha-RasV12-Induced Multilayer Cellular Aggregates are Mediated by Rac1 Activation Rather than YAP Activation”. Authors tried to reveal Rac1 function in cellular aggregation by molecular biological methods. The manuscript is well-written and the contents in this article is very important. There are a lot of work in this manuscript. However, there are some concerns to be cleared before accepting, as follows:

Major concerns:

  1. In the paragraph starting on line 289, Authors describe the importance of Rac1 in the cellular aggregate formation using Rac1-L (constitutively active form) and Rac1-N (dominant negative form). Even in Ha-RasV12 transformed cell, Rac1-N prevents cell aggregation? If so, please show the data.
  2. In figure 7, U0126 inhibited the cell aggregation in Ha-RasV12 triggered cells. Can Rac1-L is enough to initiate the cellular aggregation even in Ha-RasV12 triggered cells with U0126 treatment? Please show the data.
  3. In figure 8, no relationship between Rac1 and Rho/ROCK is shown. Please add the contribution of Rho/ROCK in this system. 

Minor concerns

  1. On lines 256 through 257, Authors describe “We already know that…”, however, there is no article cited. Please add the citation.
  2. In figure 1g, it looks there are some differences between C and I in Myosin IIA as well as in Myosin IIB. Please add p values or add “N.S.” if they are not significant.

Round 3

Reviewer 2 Report

Dear Authors,

Thank you for submitting the revision of the manuscript titled, “Ha-RasV12-Induced Multilayer Cellular Aggregates are Mediated by Rac1 Activation Rather than YAP Activation”. Authors answered major and minor concerns, however, authors did not show data in two major concerns which are critical in this manuscript. There are still major concerns to clear before accepting.

Major concerns:

  1. In the paragraph starting on line 289, Authors describe the importance of Rac1 in the cellular aggregate formation using Rac1-L (constitutively active form) and Rac1-N (dominant negative form). Even in Ha-RasV12 transformed cell, Rac1-N prevents cell aggregation? If so, please show the data.

-- In our study, we mainly used MK4 cells (MDCK cells stably transfected with

pSVlacORas and pHblacINLSneo plasmids) to investigate the mechanism of cellular aggregate formation. After MK4 cells were treated with IPTG, Ha-RasV12 expression will be triggered, which resulted in cellular aggregate formation. We overexpressed Rac1-WT, Rac1-L (constitutively active form) and Rac1-N (dominant negative form) in both MDCK cells and MK4 cells. The results showed that the activation of Rac1-L is able to induce cellular aggregate formation in both MDCK cells and MK4 cells. Since Rac1 inhibitor EHT1864 completely blocked Ha-RasV12-induced cell aggregate formation, our data indicate that augmented Rac1 activity is sufficient and required for Ha-RasV12-indcued cellular aggregate formation in MK4 cells. We didn’t test whether Rac1-N is able to prevent Ha-RasV12-induced cellular aggregate formation, but think that Rac-1N expression may do so.

---Authors should write not only their opinion but also show the data to support their rationale. This includes data that Rac1-N prevents cell aggregation even in Ha-RasV12 transformed cell.

  1. In figure 7, U0126 inhibited the cell aggregation in Ha-RasV12 triggered cells. Can Rac1-L is enough to initiate the cellular aggregation even in Ha-RasV12 triggered cells with U0126 treatment? Please show the data.

--We did not do such experiments. Without Ha-RasV12 induction, Rac1-L can induce cellular aggregate formation in MDCK and MK4 cells when p-ERK activity was not stimulated. Such conditions may be close to addition of U0126 to Ha-RasV12 triggered cells, albeit not exactly the same.

--- Authors should write not only their opinion but also show the data to support their rationale. This includes data that Rac1-L overexpression induces the cellular aggregation even in Ha-RasV12 triggered cells with U0126 treatment.

  1. In figure 8, no relationship between Rac1 and Rho/ROCK is shown. Please add the contribution of Rho/ROCK in this system.

--Thanks for the suggestion. We have added ROCK to figure 8. Please see below.

---Thank you for adding ROCK in the figure. But there is no data how ROCK is activated as well as how Rho contributes this system. Is the Rho/ROCK connected to Ras signal?

Minor concerns

  1. On lines 256 through 257, Authors describe “We already know that…”, however, there is no article cited. Please add the citation.

--We have added the reference # 22 to the sentence.

---Thank you for adding the reference. Now it is clear.

  1. In figure 1g, it looks there are some differences between C and I in Myosin IIA as well as in Myosin IIB. Please add p values or add “N.S.” if they are not significant.

--There is no difference between C and I in myosin IIA as well as myosin IIB. Yes, we have added N.S. to Fig 1g.

---Thank you for adding N.S. to the figure. However, the value of “with IPTG in Myosin IIB” in Figure 1(g) looks different from the one in the original figure. The value in the revised figure is higher than the one in the original figure (please see the attached file). If authors change the value in the figure, authors must explain why the data were changed.

Round 4

Reviewer 2 Report

Dear Auhtors,

Thank you very much for re-submitting the manuscript titled “Ha-RasV12-Induced Multilayer Cellular Aggregates are Mediated by Rac1 Activation Rather than YAP Activation”. Thank you for the comments. Authors answered all concerns to be cleared.